# Heterologous Expression of Nitrate Assimilation Related-Protein DsNAR2.1/NRT3.1 Affects Uptake of Nitrate and Ammonium in Nitrogen-Starved *Arabidopsis*

**DOI:** 10.3390/ijms21114027

**Published:** 2020-06-04

**Authors:** Hongping Ma, Junchao Zhao, Shuang Feng, Kun Qiao, Shufang Gong, Jingang Wang, Aimin Zhou

**Affiliations:** 1College of Horticulture and Landscape Architecture, Northeast Agricultural University, Harbin 150030, China; HongpingMa@aliyun.com (H.M.); ZhaoJunchaos@outlook.com (J.Z.); kunqiao@neau.edu.cn (K.Q.); shufanggong@neau.edu.cn (S.G.); 2Key Laboratory of Saline-Alkali Vegetation Ecology Restoration (Northeast Forestry University), Ministry of Education, Harbin 150040, China; shuangfeng1986@aliyun.com; 3College of Life Sciences, Northeast Forestry University, Harbin 150040, China

**Keywords:** nitrogen, *DsNRT3.1*, heterologous expression, NO_3_^−^ and NH_4_^+^ uptake, seedling growth, *Arabidopsis*

## Abstract

Nitrogen (N) is an essential macronutrient for plant growth. Plants absorb and utilize N mainly in the form of nitrate (NO_3_^−^) or ammonium (NH_4_^+^). In this study, the nitrate transporter DsNRT3.1 (also known as the nitrate assimilation-related protein DsNAR2.1) was characterized from *Dianthus spiculifolius*. A quantitative PCR (qPCR) analysis showed that the *DsNRT3.1* expression was induced by NO_3_^−^. Under N-starvation conditions, the transformed *Arabidopsis* seedlings expressing *DsNRT3.1* had longer roots and a greater fresh weight than the wild type. Subcellular localization showed that DsNRT3.1 was mainly localized to the plasma membrane in *Arabidopsis* root hair cells. Non-invasive micro-test (NMT) monitoring showed that the root hairs of N-starved transformed *Arabidopsis* seedlings had a stronger NO_3_^−^ and NH_4_^+^ influx than the wild-type seedlings, using with NO_3_^−^ or NH_4_^+^ as the sole N source; contrastingly, transformed seedlings only had a stronger NO_3_^−^ influx when NO_3_^−^ and NH_4_^+^ were present simultaneously. In addition, the qPCR analysis showed that the expression of *AtNRT2* genes (*AtNRT2.1**–2.6*), and particularly of *AtNRT2.5*, in the transformed *Arabidopsis* differed from that in the wild type. Overall, our results suggest that the heterologous expression of *DsNRT3.1* affects seedlings’ growth by enhancing the NO_3_^−^ and NH_4_^+^ uptake in N-starved *Arabidopsis*. This may be related to the differential expression of *AtNRT2* genes.

## 1. Introduction

Nitrogen (N) is an essential macronutrient for plant growth and crop productivity [1]. Plants acquire inorganic N from the soil, mainly in the form of nitrate (NO_3_^−^) and ammonium (NH_4_^+^) [2]. Under aerobic conditions, NO_3_^−^ is usually the most abundant N source in soil, and it is the predominant form of N absorbed by most plants [3]. For some species, such as rice (*Oryza sativa*) and *Camellia sinensis*, NH_4_^+^ is the preferred N source [4,5]. The NO_3_^−^ uptake system in higher plants includes a low-affinity transport system (LATS) and a high-affinity transport system (HATS) to accommodate changes in NO_3_^−^ concentrations in the soil [6].

The root uptake of NO_3_^−^ and transport to the whole plant are regulated by NO_3_^−^ transporters (NRTs) [6,7]. NRTs consists of three gene families, *NRT1* (also named NPF, nitrate transporter 1 (NRT1)/peptide transporter (PTR) gene family), *NRT2*, and *NRT3* (also named NAR, nitrate assimilation-related protein) [3]. The NRT1 family comprises many genes—53 members in *Arabidopsis* and 93 members in the rice genome [8,9]. It is suggested that most members of the NRT1 family function as the main components of the LATS at high concentrations of NO_3_^−^ [6]. Under low NO_3_^−^ concentrations, NRT2 and NRT3/NAR families can function as the major components of the HATS for the root NO_3_^−^ influx [6,7,10]. The NRT2 family has seven members (AtNRT2.1–2.7) in *Arabidopsis* [11], and five members (OsNRT2.1, 2.2, 2.3a, 2.3b, and OsNRT2.4) in the rice genome [12]. The many members of the NRT2 family are unable to transport NO_3_^−^ alone; they require a partner protein, NRT3/NAR2. The first *NRT3*/*NAR2* gene was identified in *Chlamydomonas reinhardtii*; it functions with CrNRT2 members in the NO_3_^−^ influx [13]. In both *Arabidopsis* and rice, the *NRT3*/*NAR2* families have two members, *AtNRT3.1* and *AtNRT3.2* and *OsNAR2.1* and *OsNAR2.2* [10,11,12]. *Arabidopsis* NRT2 members (AtNRT2.1–2.6) require AtNRT3.1 for NO_3_^−^ uptake [14,15]. In *a**tnrt 3.1* mutant plants, the high-affinity NO_3_^−^-inducible influx was reduced [16]. Except for AtNRT2.7, the other six AtNRT2 transporters interact with AtNRT3.1 [15]. Furthermore, the plasma membrane complex of AtNRT3.1 and AtNRT2.5 has been found to be a major contributor to constitutive high-affinity NO_3_^−^ influx in *Arabidopsis* [17]. Similarly, the rice OsNRT2 members (OsNRT2.1, 2.2 and 2.3a) require OsNAR2.1 for their NO_3_^−^ uptake [12,18]. Both high- and low-affinity NO_3_^−^ transport were greatly impaired by OsNAR2.1 knockdown. Yeast two hybrid screening showed that OsNAR2.1 interacted with OsNRT2.1, OsNRT2.2, and OsNRT2.3a [18]. Furthermore, the arginine 100 and aspartic acid 109 of OsNAR2.1 were found to be key amino acids in their interaction with OsNRT2.3a, and their interaction occurs in the plasma membrane [19]. These studies demonstrate that NRT3 is essential for high-affinity NRT2 in NO_3_^−^ uptake in *Arabidopsis* and rice.

*Dianthus spiculifolius* Schur (Caryophyllaceae), a perennial herbaceous flowering plant, is a high-economic value ornamental crop with a high economic value that has strong resistance to cold, drought, and barren conditions. Studying its tolerance toward barren soil contributes to expanding its application and production in barren soils. The *NRT* genes play key roles in N absorption and utilization by plants. In this study, the *DsNRT3.1* gene was identified from the transcriptome of *D. spiculifolius* [20]. Its expression in response to different N sources was investigated using qPCR. The subcellular localization of DsNRT3.1 in plant cells was observed using green fluorescent protein (GFP) as a marker. We compared the phenotypes of the transformed *Arabidopsis* seedlings expressing *DsNRT3.1* and wild-type seedlings with different N sources (NO_3_^−^ or NH_4_^+^). We monitored the net fluxes of NO_3_^−^ and NH_4_^+^ in transformed *Arabidopsis* root hairs with different N sources using non-invasive micro-test (NMT) technology.

## 2. Results

### 2.1. Expression Analysis of DsNRT3.1

The complete cDNA sequence of *DsNRT3.1* was isolated from the *D*. *spiculifolius* transcriptome, which contains a 585 bp open reading frame that encodes 194 amino acids and has a predicted molecular mass of 21.07 kDa. The sequence and phylogenetic tree analysis revealed that the DsNRT3.1 protein was similar to AtNRT3.1 (42.86% amino acid sequence identity) and NRT3.1 from *Arabidopsis* and other species (Figure 1A,B). Two transmembrane domains of the DsNRT3.1 protein were predicted using HMMTOP software (Figure 1C).

The expression of *DsNRT3.1* in seedlings supplied with different N sources was investigated. The qPCR analysis revealed that the addition of exogenous KNO_3_ (1 mM), KCl (1 mM), or NH_4_NO_3_ (1 mM) significantly induced the *DsNRT3.1* expression (Figure 2A,B). By 3 h of treatment, KNO_3_ and KCl induced the upregulation of *DsNRT3.1* expression by six-fold and two-fold, respectively. By 6–24 h, both KNO_3_ and KCl induced the upregulation of *DsNRT3.1* expression by 4–6-fold. By 3–24 h, the addition of NH_4_NO_3_ (1 mM) induced the upregulation of *DsNRT3.1* expression by 4–8-fold. However, (NH_4_)_2_SO_4_ (0.5 mM) did not significantly affect the *DsNRT3.1* expression (Figure 2B). This indicates that the *DsNRT3.1* expression was affected by NO_3_^−^ and K^+^.

### 2.2. Effect of Heterologous Expression of DsNRT3.1 on Arabidopsis Seedings Growth

Transformed *Arabidopsis* seedlings expressing *DsNRT3.1*, generated using the 35S promoter, were used to investigate the role of *DsNRT3.1* in plant growth. The expression of *DsNRT3.1* in the transformed *Arabidopsis* lines was confirmed using a semi-quantitative RT-PCR (Figure 3A). After 10 days on N-free 1/2 strength Murashige and Skoog (MS) medium, wild-type and transformed seedlings showed the N-starvation phenotype, but there was no significant difference in the primary root length. On N-free 1/2 MS medium supplemented with NO_3_^−^ or NH_4_^+^ (0.01 or 0.02 mM) as the sole N source, the primary roots of the transformed seedlings were significantly longer than those of the wild-type seedlings. At NO_3_^−^ or NH_4_^+^ concentrations over 1 mM, there was no significant difference in primary root length between the wild-type and transformed seedlings (Figure 3B–D). Furthermore, the fresh weight of the transformed seedlings was generally higher than that of the wild-type seedlings after 20 days on N-free 1/2 MS medium supplemented with NO_3_^−^ (0.05 to 0.5 mM) or NH_4_^+^ (0.1 mM) (Figure 4A–C). These results show that with low-concentration NO_3_^−^ or NH_4_^+^ as the sole N source, the transformed seedlings grew better than the wild-type seedlings.

### 2.3. Effect of Heterologous Expression of DsNRT3.1 on the NO_3_^−^ and NH_4_^+^ Fluxes in Arabidopsis Root Hairs

It is thought that DsNRT3.1 is a membrane protein (Figure 1C); therefore, we examined the DsNRT3.1 localization in *Arabidopsis* using GFP as a marker. In the root hair cells of *Arabidopsis* stably expressing DsNRT3.1-GFP, the DsNRT3.1-GFP signals were localized mainly to the cell periphery, which is similar to the plasma membrane. However, as a control, the GFP signal was generally distributed throughout the root hair cells of *Arabidopsis* stably expressing GFP (Figure 5A). Root hairs are a primary site for nutrient uptake in plants [21]. The net NO_3_^−^ and NH_4_^+^ fluxes in the root hairs of wild-type and transformed *Arabidopsis* were monitored and compared using NMT (Figure 5B).

Seedlings grown on the 1/2 MS medium or N-free 1/2 MS medium (N-starvation treatment) for 10 days were transferred to a test solution containing KNO_3_ (0.1 mM), NH_4_Cl (0.1 mM), or NO_3_NH_4_ (0.5 mM) as the sole N source. With KNO_3_ or NH_4_Cl as the sole N source, the root hairs of the wild-type and transformed seedlings cultured on the 1/2 MS medium displayed a marked NO_3_^−^ or NH_4_^+^ efflux, with no significant difference in the efflux rates between the transformed and wild-type seedlings (Figure 6A–D). However, there was marked NO_3_^−^ or NH_4_^+^ influx displayed in the root hairs of the N-starved wild-type and transformed seedlings (Figure 6E,G); the mean influx rate in the transformed seedlings was significantly higher than in the wild-type seedlings (Figure 6F,H).

With NO_3_NH_4_ as the sole N source, the root hairs of N-starved transformed seedlings showed a higher NO_3_^−^ influx rate than those of the wild type (Figure 7E,F), whereas the NH_4_^+^ influx rate was similar to that of the wild type (Figure 7G,H). Moreover, the NO_3_^−^ and NH_4_^+^ efflux from the root hairs of the wild-type or transformed seedlings cultured on 1/2 MS medium was not marked and showed no significant difference among the lines (Figure 7A–D). This indicates that the root hairs of N-starved transformed seedlings have a stronger NO_3_^−^ uptake ability than those of wild-type seedlings; they also have a stronger NH_4_^+^ uptake ability in the absence of NO_3_^−^, suggesting that the heterologous expression of *DsNRT3.1* affects the NO_3_^−^ and NH_4_^+^ uptake in N-starved *Arabidopsis* seedlings.

### 2.4. Effect of Heterologous Expression of DsNRT3.1 on Arabidopsis NRT2 Genes Expression

NRT3, as a partner protein to the NTR2 family, is important for NRT2 function [15]. The expression levels of *AtNRT2* genes in the wild-type and transformed *Arabidopsis* seedlings were compared. The qPCR analysis revealed that the expression levels of six of the seven *NRT2* family members differed between the transformed *Arabidopsis* and wild-type seedlings; *NRT2.1*, *2.2*, *2.4*, and *2.6* were down-regulated and *NRT2.5* was up-regulated. However, the expression of *AtNRT3.1* and *AtNRT2.7* in transformed *Arabidopsis* was similar to that in the wild type, with no significant difference (Figure 8). This suggests that the heterologous expression of *DsNRT3.1* affects the expression of members of the *AtNRT2* gene family in transformed *Arabidopsis* seedlings.

## 3. Discussion

The first NRT3.1 was identified from *Chlamydomonas* [13], and its orthologous proteins were subsequently identified in barley (HvNRT3.1/HvNAR2.1) [22], *Arabidopsis* (AtNRT3.1) [16], rice (OsNRT3.1/OsNAR2.1) [18], and chrysanthemums (CmNRT3/CmNAR2) [23]. A sequence analysis showed that DsNRT3.1 was highly similar to AtNRT3.1, HvNRT3.1, and OsNRT3.1/OsNAR2.1 (Figure 1A,B). NRT3.1 was considered to be a high-affinity transporter; it was induced at low concentrations of N sources, whereas its expression was largely unaffected at high concentrations or using saturated N sources. Therefore, we performed a gene expression analysis after N-starvation. In N-starved seedlings, *DsNRT3.1* was also induced by NO_3_^−^ (Figure 2). Similarly, the expression of *AtNRT3.1*, *OsNRT3.1/OsNAR2.1*, and *CmNRT3/CmNAR2* was also induced by NO_3_^−^ in N-starved seedlings [16,18,23]. However, the expression of *OsNRT3.1/OsNAR2.1* was almost unaffected by NH_4_^+^ [18]. We found that *DsNRT3.1* expression was not induced by NH_4_^+^ (Figure 2B). In addition, K^+^ affected the *DsNRT3.1* expression (Figure 2A). NRT1.5 from the NRT family serves as a proton-coupled H^+^/K^+^ antiporter for K^+^ loading in *Arabidopsis* [24].

The roots of N-starved transformed *DsNRT3.1* seedlings were longer than those of the wild type under low N conditions, with NO_3_^−^ or NH_4_^+^ as the sole N source (Figure 3). After longer periods of culture, the fresh weight of the transformed *DsNRT3.1* seedlings was generally higher than that of the wild type (Figure 4). With KNO_3_ as the sole N source, the transformed seedlings had longer roots and higher root NO_3_^−^ influx rates than the wild type (Figure 3C and Figure 6E,F). Likewise, with NH_4_NO_3_ as the sole N source, the transformed seedling roots had higher NO_3_^−^ influx rates than the wild type (Figure 7E,F). These results suggest that longer root growth in transformed seedlings may be associated with a stronger NO_3_^−^ uptake. A confocal observation using GFP as a marker revealed that DsNRT3.1 was mainly localized to the plasma membrane in *Arabidopsis* root hair cells (Figure 5A). Furthermore, NMT monitoring showed that the root hairs of N-starved transformed seedlings had a stronger NO_3_^−^ and NH_4_^+^ influx than those of the wild-type seedlings (Figure 6A–D). Several studies have shown that NRT3.1 regulates NO_3_^−^ uptake via interaction with NRT2 members on the plasma membrane [15,18,19,23]. Thus, we hypothesized that the heterologous expression of *DsNRT3.1* might affect the NO_3_^−^ uptake in N-starved seedlings via interaction with AtNRT2 members on the plasma membrane. We compared the expression levels of *AtNRT2* genes in the transformed and wild-type seedlings. The qPCR analysis revealed that the expression of multiple *NRT2* members (*AtNRT2.1–2.6*), and particularly of *AtNRT2.5*, was altered in transformed *Arabidopsis*-expressing *DsNRT3.1* (Figure 8). *Arabidopsis* AtNRT2.5 is a high-affinity NO_3_^−^ transporter localized on the plasma membrane and is expressed in the root hair zone [25]. Under long-term N-starvation, *AtNRT2.5* becomes the most abundant transcript amongst the seven *AtNRT2* members in *Arabidopsis* shoots and roots [25]. Subsequent studies have revealed that AtNRT3.1 and AtNRT2.5 form a complex on the plasma membrane; this complex is a major contributor to high-affinity NO_3_^−^ influx in *Arabidopsis* [17]. However, in our study, the *AtNRT3.1* expression in transformed seedlings was not altered (Figure 8). Thus, our results suggest that the heterologous expression of *DsNRT3.1* may enhance the NO_3_^−^ uptake by affecting *AtNRT2.5* function, thereby affecting the seedling growth in *Arabidopsis* under N-starvation. To further elucidate the role and function of DsNRT3.1, future research should address the following aspects: (i) whether the heterologous expression of *DsNRT3.1* complements the function of the *atnrt3.1* mutant; and (ii) whether DsNRT3.1 interacts with AtNRT2 members, and particularly with AtNRT2.5, on the plasma membrane. Furthermore, we found that in the absence of NO_3_^−^, the root hairs of N-starved transformed *Arabidopsis* seedlings showed a stronger NH_4_^+^ influx than those of the wild type (Figure 6G,H and Figure 7G,H), suggesting that the heterologous expression of *DsNRT3.1* may directly or indirectly affect the NH_4_^+^ uptake under N-starvation. However, this requires further study. Recent studies have shown that AtNRT1.1 participates in the NH_4_^+^ uptake by affecting the NH_4_^+^ transporter expression [26].

## 4. Materials and Methods

### 4.1. Sequence Analysis of DsNRT3.1

The amino acid sequence of DsNRT3.1 (GenBank accession No. MN334698) and its homologs were aligned using BioEdit (https://bioedit.software.informer.com/). The phylogenetic tree was constructed using molecular evolutionary genetics analysis (MEGA) 4.1 software (available online: http://www.megasoftware.net/). The transmembrane domains prediction of DsNRT3.1 was performed using the HMMTOP (http://www.sacs.ucsf.edu/cgi-bin/hmmtop.py/) software.

### 4.2. Plant Materials and Growth Conditions

Seeds of *D. spiculifolius* and *A. thaliana* (Columbia-0) were surface sterilized using 30% (*v*/*v*) bleach solution for 6 min and rinsed five times with sterile water. The sterilized seeds were grown on 1/2 strength Murashige and Skoog (MS) medium (10.3 mM NH_4_NO_3_, 9.4 mM KNO_3_, 0.6 mM KH_2_PO_4_, 0.8 mM MgSO_4_, 1.5 mM CaCl_2_, 2.5 µM KI, 50 µM MnSO_4_, 50 µM H_3_BO_3_, 15 µM ZnSO_4_, 0.5 µM Na_2_MoO_4_, 0.05 µM CuSO_4_, 0.06 µM CoCl_2_, 0.1 mM FeSO_4_, 0.1 mM Na_2_-EDTA) medium supplemented with 1% agar and 3% sucrose (pH 5.8) under a 12 h light/12 h dark photoperiod (100 μmol m^−2^ s^−1^ light intensity) at 22 °C.

For the different N-source treatments, *D. spiculifolius* seedings were N-starved for 1 week, then transferred to N-free 1/2 MS medium (3% sucrose, 1% agar, pH 5.8) supplemented with 1 mM KNO_3_, 1 mM KCl, 1 mM NH_4_NO_3_, or 0.5 mM (NH_4_)_2_SO_4_ as the sole N source. At least 10 seedlings from each treatment were harvested and pooled at different time points (0, 3, 6, 12, or 24 h after treatments), frozen immediately in liquid N, and stored at −80 °C for RNA preparation.

### 4.3. RNA Preparation and Expression Analysis

The total RNA (1.2 µg) was extracted using an RNeasy^®^ Mini Kit (Qiagen, Valencia, CA, USA), and cDNA (20 µL) was synthesized using an M-MLV RTase cDNA Synthesis Kit (TaKaRa, Shiga, Japan), according to the manufacturers’ instructions. The qPCR analysis was performed using Bio-Rad CFX Manager Software Version 3.1 (Bio-Rad, Hercules, CA, USA) using the appropriate pairs of species-specific primers (Appendix A). The reaction components per 20 µL were as follows: 7 µL H_2_O, 10 µL Hieff qPCR SYBR Green Master Mix (YEASEN, Shanghai, China), 1 µL 10 µM of each primer and 1 µL cDNA. The thermal cycling program was as follows: initial denaturation at 95 °C for 60 s, and 40 cycles at 95 °C for 10 s, 60 °C for 30 s, and 72 °C for 30 s. *DsActin* or *AtActin* was used as an internal reference gene. The relative quantification of gene expression was evaluated using the delta-delta-Ct method. Three biological replicates and three technical replicates were performed for each analysis.

### 4.4. Vector Construction and Plant Transformation

The open reading frame of *DsNRT3.1* was constructed into the pBI121 vector driven by 35S promoter, using the following restriction sites: 121DsNRT3.1(X)-F (TCTAGAATGGCGGTGCGAGG ATTAAC; the XbaI site is underlined) and 121DsNRT3.1(S)-R (GAGCTCGTAGCTAGCTTCGACTT CTTC; the SacI site is underlined). To construct the DsNRT3.1-GFP fusion gene, the open reading frame of DsNRT3.1 without a stop codon was constructed into a pBI121-GFP vector using the 121DsNRT3.1(X)-F and 121DsNRT3.1(K)-R (GGTACCGTAGCTAGCTTCGACTTCTTC; the KpnI site is underlined) restriction sites. These pBI121-GFP constructs have been described elsewhere [27]. These constructs were transformed into the *Agrobacterium tumefaciens* strain EHA105 for plant transformation; *Arabidopsis* was transformed using the floral dip method [28]. The transformed plants were placed on 1/2 MS medium containing 30 μg mL^−1^ kanamycin. The primers used in this study are shown in Appendix A.

### 4.5. Subcellular Localization of DsNRT3.1

Four-day-old seedlings grown on vertical 1/2 MS medium plates (0.5% sucrose, pH 5.8) were placed at room temperature. The roots of transformed *Arabidopsis* seedlings were re-washed using liquid 1/2 MS medium immediately before confocal laser scanning microscopy (CLSM) (Nikon, A1, Tokyo, Japan).

### 4.6. Phenotypic Analysis of Transformed Arabidopsis

The sterilized wild-type and transformed *Arabidopsis* seeds were stratified for 3 days at 4 °C. The seedlings were then transferred to N-free 1/2 MS medium (3% sucrose, 1% agar, pH 5.8) supplemented with different concentrations of KNO_3_ or NH_4_Cl (0, 0.01, 0.02, 0.05, 0.1, 0.3, 0.5, 1, or 2.5 mM) for 10 days before measuring root length, or 20 days before measuring the seedlings fresh weight. After examining seedling growth phenotypes, the plants were photographed using a Canon EOS 80D camera (Canon, Tokyo, Japan). Images were processed using Adobe Photoshop CS. Seedling root length was measured using Image Pro Plus 6.0 software (Media Cybernetics, Silver Spring, MD, USA). The fresh weight of the aerial parts of the seedlings was measured using an analytical balance with an accuracy of ± 0.1 mg.

### 4.7. NMT Measurement of Net NO_3_^−^ and NH_4_^+^ Fluxes in Arabidopsis Root Hairs

The net fluxes of NO_3_^−^ and NH_4_^+^ in the root hairs of the wild-type and transformed *Arabidopsis* seedlings were measured using NMT (NMT100 Series, YoungerUSA LLC, Amherst, MA, USA) as previously described [29]. The wild-type and transformed *Arabidopsis* seeds were surface sterilized and kept at 4 °C for 3 days in the dark. After germination, the seedlings were transferred into N-free 1/2 MS medium for 7 days, after which the roots of the seedlings were immediately equilibrated in a measuring solution containing 0.1 mM KNO_3_, 0.1 mM NH_4_Cl, or 0.1 mM NH_4_NO_3_ for 10 min. The roots were fixed to the bottom of the plate using resin blocks and filter paper strips. To take flux measurements, the ion-selective electrodes were calibrated using NO_3_^−^ or NH_4_^+^ at concentrations of 0.05 and 0.5 mM, respectively. The net fluxes of NO_3_^−^ or NH_4_^+^ in the root hairs were measured. Each sample was measured continuously for 10 min. The KNO_3_ and NH_4_NO_3_ measuring solution contained 0.1 mM KNO_3_, 0.1 mM CaCl_2_, 0.1 mM KCl, and 0.3 mM MES (pH 6.0). The NH_4_Cl measuring solution contained 0.1 mM NH_4_Cl, 0.1 mM CaCl_2_, and 0.3 mM MES (pH 6.0). The KNO_3_ calibration solution contained 0.5mM KNO_3_, 0.1 mM KCl, 0.1 mM CaCl_2_ and 0.3 mM MES (pH 6.0); 0.05 mM KNO_3_, 0.1 mM KCl, 0.1 mM CaCl_2_, and 0.3 mM MES (pH 6.0). The NH_4_Cl calibration solution contained 0.5 mM NH_4_Cl, 0.1 mM CaCl_2_, and 0.3 mM MES (pH 6.0); 0.05 mM NH_4_Cl, 0.1 mM CaCl_2_, and 0.3 mM MES (pH 6.0). The NH_4_NO_3_ calibration solution contained 0.5 mM NH_4_NO_3_, 0.1 mM CaCl_2_, and 0.3 mM MES (pH 6.0); 0.05 mM NH_4_NO_3_, 0.1 mM CaCl_2_, and 0.3 mM MES (pH 6.0). The calibration slopes for NO_3_^−^ and NH_4_^+^ were −55.61 and 56.19 mV/decade, respectively. Six biological repeats were performed for each analysis.

### 4.8. Statistical Analysis

The data were analyzed using a one-way analysis of variance in SPSS (SPSS, Inc., Chicago, IL, USA), and statistically significant differences were calculated using the Student’s *t*-test, with *p* < 0.05 as the threshold for significance.

## 5. Conclusions

In this study, the nitrate assimilation-related protein DsNAR2.1/DsNRT3.1 was characterized from *D. spiculifolius*. DsNRT3.1 was localized mainly in the plasma membrane of *Arabidopsis* root hair cells. The findings reveal that the heterologous expression of *DsNRT3.1* in *Arabidopsis* affects seedling root growth and NO_3_^−^ and NH_4_^+^ uptake under N-starvation, as well as the expression of *AtNRT2* family members. Overall, our results suggest that DsNRT3.1 may affect the seedling growth in N-starved *Arabidopsis* by enhancing the NO_3_^−^ and NH_4_^+^ uptake, which may be associated with altered *AtNRT2* family expression. Our study also suggests that DsNRT3.1 functions as a positive regulator of plant N uptake and could be utilized for the genetic improvement of N-efficient plants.

## Figures and Tables

**Figure 1 ijms-21-04027-f001:**
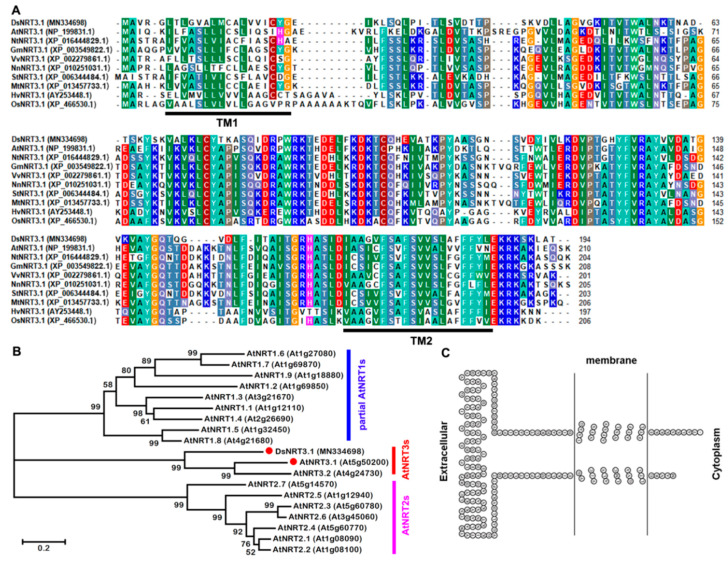
Sequence analysis of DsNRT3.1. (**A**) Amino acid sequence alignment of NRT3.1 from multiple plant species. The black lines indicate the positions of the two transmembrane (TM) domains. (**B**) Phylogenetic tree of DsNRT3.1 with NRT family members (AtNRT1, AtNRT2, and AtNRT3) from *Arabidopsis*. (**C**) Transmembrane domain prediction of DsNRT3.1.

**Figure 2 ijms-21-04027-f002:**
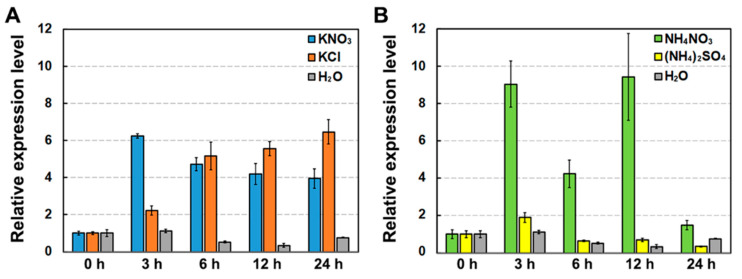
Effects of nitrate and ammonium nitrogen on the *DsNRT3.1* gene expression. One-week-old *D. spiculifolius* seedlings were transferred to a H_2_O solution (negative control), a H_2_O solution containing 1 mM of KNO_3_ or 1 mM of KCl (**A**), or a H_2_O solution containing 1 mM of NH_4_NO_3_ or 0.5 mM of (NH_4_)_2_SO_4_ (**B**) treated for 0, 3, 6, 12, and 24 h. *DsActin* was used as an internal reference. Error bars represent the standard error (*n* = 3).

**Figure 3 ijms-21-04027-f003:**
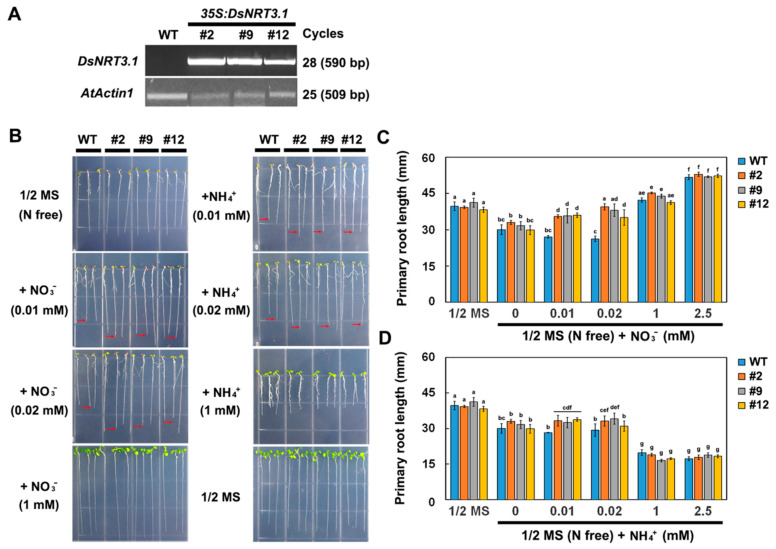
Comparison of the seedling root growth of wild-type (WT) and transformed *Arabidopsis* at different N supply levels. (**A**) Semi-quantitative RT-PCR analysis of the *DsNRT3.1* expression in WT and transformed *Arabidopsis* lines (#2, #9, and #12). Phenotype (**B**) and root length (**C**,**D**) of WT and transformed seedlings grown on 1/2 Murashige and Skoog (MS) medium or N-free 1/2 MS medium supplemented with different concentrations (0, 0.01, 0.02, 1, and 2.5 mM) of KNO_3_ or NH_4_Cl for 10 days. Different letters indicate significant differences (Student’s *t*-test; *p* < 0.05). Error bars represent the standard error (*n* = 6).

**Figure 4 ijms-21-04027-f004:**
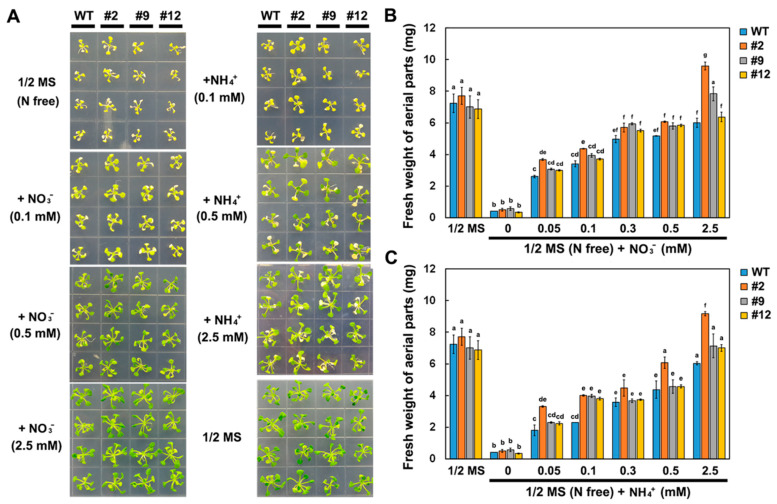
Comparison of seedling fresh weight of wild-type (WT) and transformed *Arabidopsis* at different N supply levels. Phenotype (**A**) and fresh weight (**B**,**C**) of WT and transformed seedlings grown on 1/2 MS medium or N-free 1/2 MS medium supplemented with different concentrations (0, 0.05, 0.1, 0.3, 0.5, and 2.5 mM) of KNO_3_ or NH_4_Cl for 20 days. Different letters indicate significant differences (Student’s *t*-test; *p* < 0.05). Error bars represent the standard error (*n* = 6).

**Figure 5 ijms-21-04027-f005:**
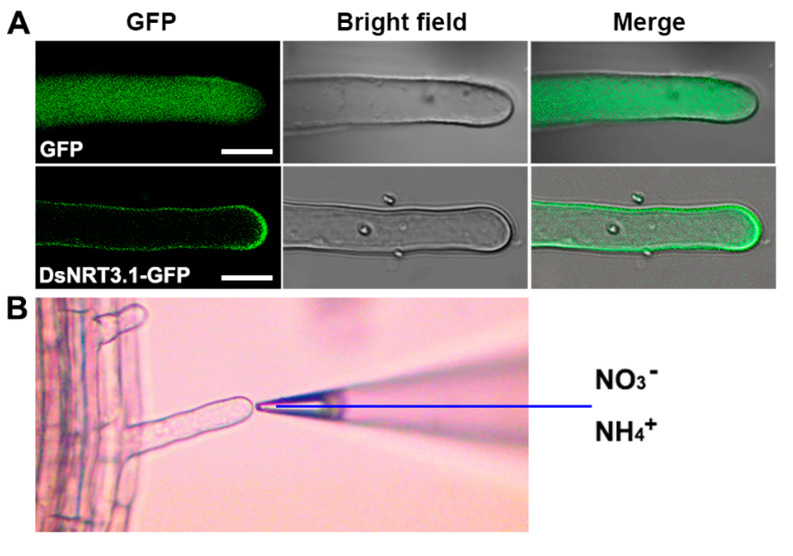
Subcellular localization of DsNRT3.1-green fluorescent protein (GFP) in *Arabidopsis* root hair cells. (**A**) Confocal images of an *Arabidopsis* root hair cell stably expressing GFP or DsNRT3.1-GFP. GFP fluorescence is green. Merged images were created by merging the GFP and bright field images. Scale bars = 10 µm. (**B**) Morphology and root hair cell sites were monitored using a non-invasive micro-test (NMT). Ion (NO_3_^−^ or NH_4_^+^) selective microelectrodes were used. The monitoring distances were 0 and 20 µm from the end of the root hair cells. The ion flux rate, based on the voltages monitored between two points (0 and 20 µm), was calculated using iFluxes/imFluxes 1.0 software (Younger USA LLC, Amherst, MA, USA).

**Figure 6 ijms-21-04027-f006:**
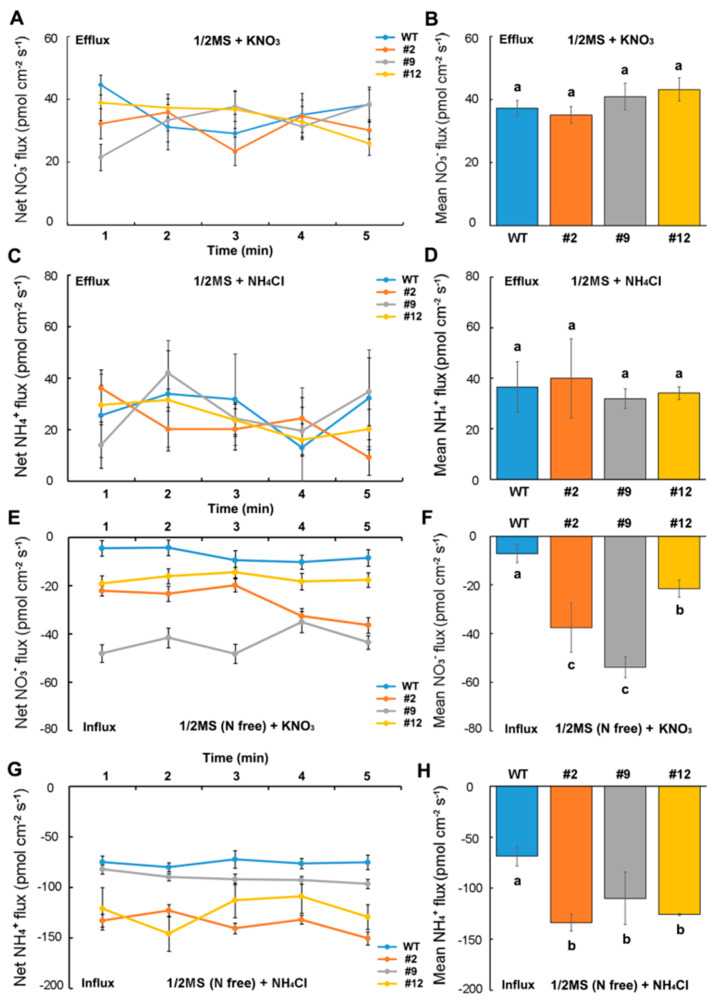
NO_3_^−^ and NH_4_^+^ flux rates in the root hairs of wild-type (WT) and transformed *Arabidopsis* supplied with KNO_3_ or NH_4_Cl as the sole N source. WT and transformed seedlings were grown on 1/2 MS medium or N-free 1/2 MS medium for 10 days and transferred to a test solution containing 0.1 mM KNO_3_ or NH_4_Cl for 10 min. Then, the net NO_3_^−^ and NH_4_^+^ fluxes were monitored. Net and mean NO_3_^−^ (**A**,**B**) or NH_4_^+^ (**C**,**D**) flux rates of the WT and transformed seedlings grown on the 1/2 MS medium. Net and mean NO_3_^−^ (**E**,**F**) or NH_4_^+^ (**G**,**H**) flux rates of the WT and transformed seedlings grown on the N-free 1/2 MS medium. Mean NO_3_^−^ (**B**,**F**) or NH_4_^+^ (**D**,**H**) flux rates of six samples (*n* = 6). Different letters indicate significant differences among the transformed lines and WT seedlings (Student’s *t*-test; *p* < 0.05). Error bars represent the standard error.

**Figure 7 ijms-21-04027-f007:**
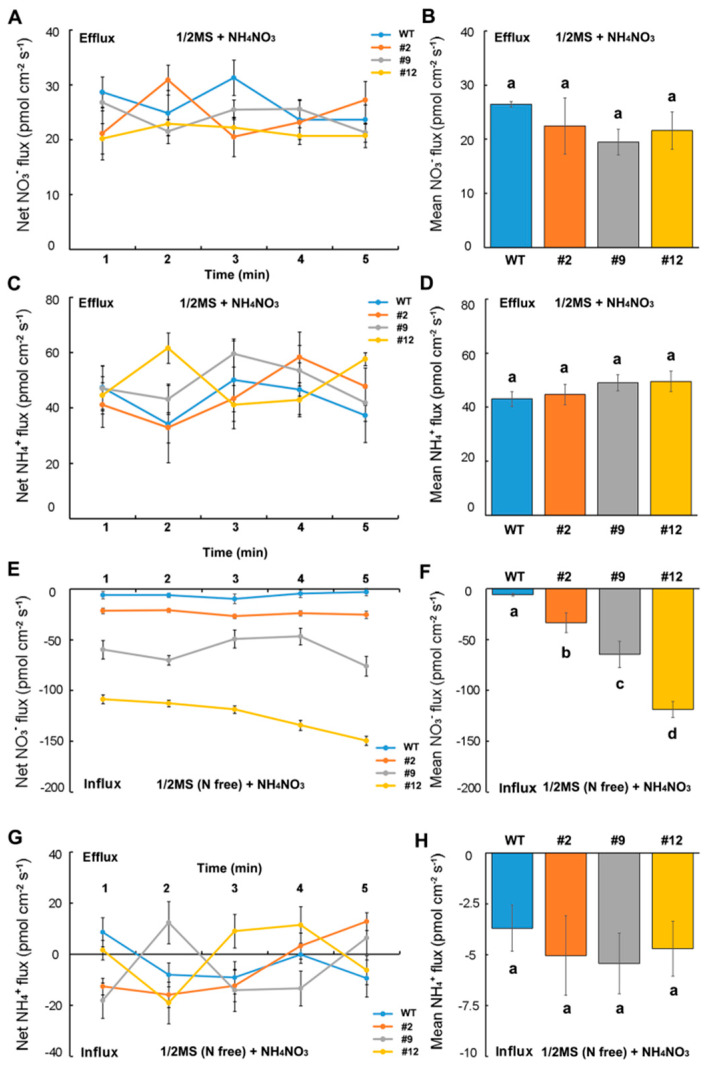
NO_3_^−^ and NH_4_^+^ flux rates in the root hairs of wild-type (WT) and transformed *Arabidopsis* with NO_3_NH_4_ as the sole N source. The WT and transformed seedlings grown on 1/2 MS medium or N-free 1/2 MS medium for 10 d were transferred to a test solution containing 0.1 mM NO_3_NH_4_ for 10 min, and the net NO_3_^−^ or NH_4_^+^ flux was monitored. Net and mean NO_3_^−^ (**A**,**B**) or NH_4_^+^ (**C**,**D**) flux rates of the WT and transformed seedlings grown on the 1/2 MS medium. Net and mean NO_3_^−^ (**E**,**F**) or NH_4_^+^ (**G**,**H**) flux rates of the WT and transformed seedlings grown on the N-free 1/2 MS medium. Mean NO_3_^−^ (**B**,**F**) or NH_4_^+^ (**D**,**H**) flux rates of six samples (*n* = 6). Different letters indicate significant differences among the transformed lines and WT seedlings (Student’s *t*-test; *p* < 0.05). Error bars represent the standard error.

**Figure 8 ijms-21-04027-f008:**
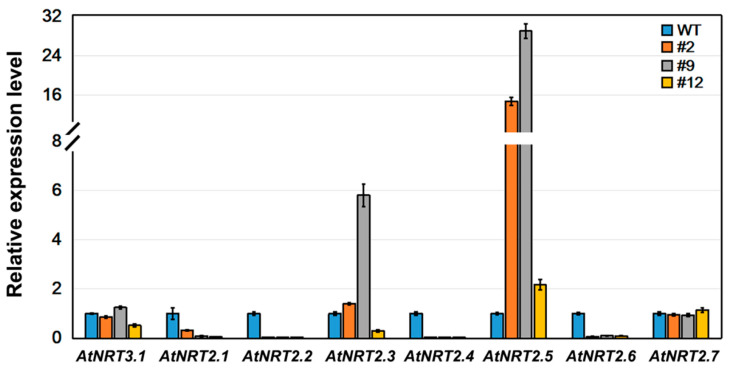
Expression analysis of *AtNRT3.1* and *AtNRT2* gene families in wild-type and transformed *Arabidopsis*. *AtActin**2* was used as an internal reference. The transcript level of wild-type (WT) seedlings was set at 1.0. Error bars represent the standard error (*n* = 3).

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
