# Peer review of "Heterologous Expression of Nitrate Assimilation Related-Protein DsNAR2.1/NRT3.1 Affects Uptake of Nitrate and Ammonium in Nitrogen-Starved Arabidopsis"

_ijms, 2020, doi:10.3390/ijms21114027_

Round 1

Reviewer 1 Report

The paper on heterologous expression of NRT3.1 from D.spiculifolius in Arabidopsis presents clear results of the gene's role in nitrogen uptake.

Some remarks:

I am not completely happy about the statistical analysis being carried out as paired t-tests, this overweights the WT which is used in each comparison. A global analysis should be carried out which takes into account the overall variability and samples classed as 'a' 'b' 'ab' etc.

Also the Materials and methods section contains nothing on the statistical analysis.

The legend of Figure 2 does not correspond to the Figure which has only parts A and B. It would be good to use extra colour to distinguish the experimental conditions - the same colours are used in A and B for different N sources.

Author Response

Response to the reviewers’ comments

Open Review #1

Comments and Suggestions for Authors

The paper on heterologous expression of NRT3.1 from D.spiculifolius in Arabidopsis presents clear results of the gene's role in nitrogen uptake.

Some remarks:

1. Comment:I am not completely happy about the statistical analysis being carried out as paired t-tests, this overweights the WT which is used in each comparison. A global analysis should be carried out which takes into account the overall variability and samples classed as 'a' 'b' 'ab' etc.

Response: Thank you for your comment. We have changed the style of statistical analysis of the data in Figure 3, 4, 6 and 7. (Pages 4, 5, 9, and 10; lines 127, 134, 173, and 195).

2. Comment:Also the Materials and methods section contains nothing on the statistical analysis.

Response: We have added statistical analysis to materials and methods. (Page 15; lines 358-361).

3. Comment:The legend of Figure 2 does not correspond to the Figure which has only parts A and B. It would be good to use extra colour to distinguish the experimental conditions - the same colours are used in A and B for different N sources.

Response: Thank you for your comment. We have changed Figure 2 and its legend. (Page 3; line 106).

Reviewer 2 Report

This study the nitrate transporter DsNRT3.1 was characterized from Dianthus spiculifolius. Subcellular localization showed DsNRT3.1 was mainly localized to the plasma membrane in Arabidopsis root hair cells. The qPCR analysis showed that the expression of AtNRT2 genes (AtNRT2.1 to AtNRT2.6), especially AtNRT2.5, in the transformed Arabidopsis was altered compared to that in the wild-type seedlings. Although the overall interest and visibility of this work, some aspects should still be considered to improve the quality and objectiveness of this work. Overall, it is an important study, and should be considered for publication, once the issues have been resolved.

  • Background of the study should be made to very clear. Provide more details of introduction and review of the work.
  • Please speculate about the reasons to the obtained results.
  • Figures: all the figure qualities should be improved.
  • In Conclusion, authors should add significance of this research to potential practical application.
  • The whole MS need to be improved.
  • Lack of results. If possible, author needs to add some experiments. For example Plant Protein Isolation and Immunoblot Analysis etc.…
  • English writing needs to be improved.

Author Response

Open Review #2

Comments and Suggestions for Authors

This study the nitrate transporter DsNRT3.1 was characterized from Dianthus spiculifolius. Subcellular localization showed DsNRT3.1 was mainly localized to the plasma membrane in Arabidopsis root hair cells. The qPCR analysis showed that the expression of AtNRT2 genes (AtNRT2.1 to AtNRT2.6), especially AtNRT2.5, in the transformed Arabidopsis was altered compared to that in the wild-type seedlings. Although the overall interest and visibility of this work, some aspects should still be considered to improve the quality and objectiveness of this work. Overall, it is an important study, and should be considered for publication, once the issues have been resolved.

1. Comment:Background of the study should be made to very clear. Provide more details of introduction and review of the work.

Response: Thank you for your comment. We have added more details about NRT3 study in the introduction. (Page, ; lines ).

 2. Comment:Please speculate about the reasons to the obtained results.

Response: We have added speculation on the reasons for the obtained results in the Discussion section. (Page 2; lines 58-70).

 3. Comment:Figures: all the figure qualities should be improved.

Response: We have improved Figure 2, 3, 4, 5, 6, 7, and their legend. (please see revised manuscript).

4. Comment:In Conclusion, authors should add significance of this research to potential practical application.

Response: We have added content about the significance of our study in the Conclusion section. (Page 16; lines 370-371).

“Our study also suggests that DsNRT3.1 functions as a positive regulator of plant N uptake that could be utilized for the genetic improvement of N-efficient plants. ”

5. Comment:The whole MS need to be improved.

Response: We have revised the whole manuscript. (please see revised manuscript).

6. Comment:Lack of results. If possible, author needs to add some experiments. For example Plant Protein Isolation and Immunoblot Analysis etc.…

Response: Thank you for your comment. We have planned to use FM4-64 dye as a plasma membrane marker to color Arabidopsis root cells to confirm DsNRT3-GFP localization. Furthermore, we also planned to investigate whether DsNRT3.1 interact with AtNRT2 members using BiFC test. We hope to publish more detailed results at a later stage to further support the findings of this study.

7. Comment:English writing needs to be improved.

Response: We re-edited the manuscript for the english language.

Other:

1) We have revised the references. (Page 17; lines 419-446).

2) We corrected the co-author's email. (Page 1; lines 12-13).

Round 2

Reviewer 2 Report

Requested corrections were completed.